# Molecular Characterization of the Enterohemolysin Gene (*ehxA*) in Clinical Shiga Toxin-Producing *Escherichia coli* Isolates

**DOI:** 10.3390/toxins13010071

**Published:** 2021-01-19

**Authors:** Ying Hua, Ji Zhang, Cecilia Jernberg, Milan Chromek, Sverker Hansson, Anne Frykman, Yanwen Xiong, Chengsong Wan, Andreas Matussek, Xiangning Bai

**Affiliations:** 1Department of Microbiology, School of Public Health, Southern Medical University, Guangzhou 510515, China; hying615@smu.edu.cn (Y.H.); gzwcs@smu.edu.cn (C.W.); 2Department of Laboratory Medicine, Division of Clinical Microbiology, Karolinska Institutet, 141 52 Stockholm, Sweden; 3Molecular Epidemiology and Public Health Laboratory, School of Veterinary Sciences, Massey University, Palmerston North 4100, New Zealand; jizhang.palmy@gmail.com; 4The Public Health Agency of Sweden, 171 82 Solna, Sweden; cecilia.jernberg@folkhalsomyndigheten.se; 5Division of Pediatrics, Department of Clinical Science, Intervention and Technology, Karolinska Institutet and Karolinska University Hospital, 141 86 Stockholm, Sweden; milan.chromek@sll.se; 6Queen Silvia Children’s Hospital, Sahlgrenska University Hospital, 413 45 Gothenburg, Sweden; sverker.hansson@pediat.gu.se (S.H.); anne.frykman@vgregion.se (A.F.); 7Department of Pediatrics, Department of Pediatrics, Sahlgrenska Academy, Sahlgrenska Academy, University of Gothenburg, 416 85 Gothenburg, Sweden; 8State Key Laboratory of Infectious Disease Prevention and Control, National Institute for Communicable Disease Control and Prevention, Chinese Center for Disease Control and Prevention, Beijing 102206, China; xiongyanwen@icdc.cn; 9Laboratory Medicine, Jönköping Region County, Department of Clinical and Experimental Medicine, Linköping University, 551 85 Jönköping, Sweden; 10Oslo University Hospital, 0372 Oslo, Norway; 11Division of Laboratory Medicine, Institute of Clinical Medicine, University of Oslo, 0372 Oslo, Norway; 12Division of Infectious Diseases, Department of Medicine Huddinge, Karolinska Institutet, 141 86 Stockholm, Sweden

**Keywords:** Shiga toxin-producing *Escherichia coli*, enterohemolysin, *ehxA*, gene diversity, hemolytic uremic syndrome, clinical significance

## Abstract

Shiga toxin (Stx)-producing *Escherichia coli* (STEC) is an important foodborne pathogen with the ability to cause bloody diarrhea (BD) and hemolytic uremic syndrome (HUS). Little is known about enterohemolysin-encoded by *ehxA*. Here we investigated the prevalence and diversity of *ehxA* in 239 STEC isolates from human clinical samples. In total, 199 out of 239 isolates (83.26%) were *ehxA* positive, and *ehxA* was significantly overrepresented in isolates carrying *stx*_2a_ + *stx*_2c_ (*p* < 0.001) and *eae* (*p* < 0.001). The presence of *ehxA* was significantly associated with BD and serotype O157:H7. Five *ehxA* subtypes were identified, among which, *ehxA* subtypes B, C, and F were overrepresented in *eae*-positive isolates. All O157:H7 isolates carried *ehxA* subtype B, which was related to BD and HUS. Three *ehxA* groups were observed in the phylogenetic analysis, namely, group Ⅰ (*ehxA* subtype A), group Ⅱ (*ehxA* subtype B, C, and F), and group Ⅲ (*ehxA* subtype D). Most BD- and HUS-associated isolates were clustered into *ehxA* group Ⅱ, while *ehxA* group Ⅰ was associated with non-bloody stool and individuals ≥10 years of age. The presence of *ehxA* + *eae* and *ehxA* + *eae + stx*_2_ was significantly associated with HUS and O157:H7 isolates. In summary, this study showed a high prevalence and the considerable genetic diversity of *ehxA* among clinical STEC isolates. The *ehxA* genotypes (subtype B and phylogenetic group Ⅱ) could be used as risk predictors, as they were associated with severe clinical symptoms, such as BD and HUS. Furthermore, *ehxA*, together with *stx* and *eae,* can be used as a risk predictor for HUS in STEC infections.

## 1. Introduction

Shiga toxin-producing *Escherichia coli* (STEC) is an important enteric foodborne pathogen that can cause bloody diarrhea (BD), hemorrhagic colitis, and potentially fatal hemolytic uremic syndrome (HUS) in infected humans [1]. STEC is estimated to cause 2.8 million cases of enteric disease in humans per year globally [2]. Over 400 STEC serotypes have been identified, among which, O157:H7 is the most prevalent serotype and is linked to severe human illness, such as HUS [3,4]. Nevertheless, since the early 2010s, non-O157 STEC, especially the so-called “top six” serogroups (O26, O45, O103, O111, O121, and O145), have been associated with continuously increasing numbers of STEC outbreaks and may account for up to 80% of STEC infections [2,5,6]. Humans are infected through contact with infected animals or the consumption of STEC-contaminated water, vegetables, milk, or meat.

Shiga toxins (Stx_1_ and Stx_2_) are the main virulence factors of STEC, which can mediate a significant cytotoxic effect in human vascular endothelial cells [7]. *stx* genes are located in the genomes of Shiga-toxin-converting bacteriophages [8]. At least 15 *stx*_1_ and *stx*_2_ gene subtypes have been identified, among which, *stx*_2a_, *stx*_2c_, and *stx*_2d_ are more virulent than other subtypes as they are highly associated with severe clinical outcomes, such as HUS [9]. Besides Stx, *eae*-encoding intimin, which is responsible for the intimate adherence of STEC, is also a significant virulence trait of pathogenic STEC [10,11]. In addition, hemolysin-encoding genes have been regarded as STEC virulence markers [7]. So far, four different types of hemolysins have been identified, namely, alpha-hemolysin (*hlyA*), silent hemolysin (*sheA*), bacteriophage-associated enterohemolysin (*e-hlyA*), and plasmid-carried enterohemolysin (*ehxA*) [12]. Enterohemolysin displays hemolytic activity that enables STEC to be observed on washed sheep blood agar, which is commonly used as a phenotypic indicator of STEC strains [13,14]. It is noteworthy that *ehxA* is prevalent in STEC strains and is closely associated with isolates causing diarrheal disease and HUS [15].

Enterohemolysin belongs to the repeats in toxin (RTX) family, which has a pore-forming capacity [16]. *ehxA* is located on a large virulence plasmid and its nucleic acid sequence has about 3000 base pairs [17]. The presence of *ehxA* has a close association with *stx*, thus it is proposed as an epidemiological marker for the rapid characterization of STEC strains [13]. For example, in the U.S. Food and Drug Administration *E. coli* Identification (FDA-ECID) microarray, *ehxA* is included as one of the genetic markers for the rapid characterization of STEC isolates [18]. Six genetically distinct *ehxA* subtypes (A to F) have been described in *E. coli* by using PCR in combination with restriction fragment length polymorphism (RFLP) analysis [12]. STEC *ehxA* subtypes differ significantly among strains isolated from different sources. Subtypes A and C are mostly found in animal isolates, where subtype A is detected in food-associated strains and subtype C is commonly found in clinical strains [12]. However, such data are limited; the correlation between *ehxA* subtypes and strains sources, as well as the clinical relevance, remains to be further elucidated.

A recent study in Sweden showed that almost all of the *eae*-positive isolates, except one, harbored *ehxA*, and the coexistence of *ehxA* and *eae* was shown to be associated with BD [19]. In a previous study, only 10.9% of isolates carried *eae* among 138 *ehxA*-positive non-O157 STEC isolates from human, animal, and food sources in China, and 61.54% of these were clinically relevant [20]. The aim of this study was to investigate the prevalence and genetic diversity of the *ehxA* gene, its correlation with serotypes, and the presence of *stx* and *eae*. Furthermore, we aimed to assess the association between *ehxA* subtypes and disease severity.

## 2. Results

### 2.1. Distribution of ehxA in the Clinical STEC Isolates

Among the 239 STEC strains isolated in Sweden, *ehxA* was identified in 199 (83.26%) isolates. Fifty-three (26.63%) of the *ehxA*-positive isolates were from patients with HUS, 47 (23.62%) were from patients with BD, and 99 (49.75%) were from individuals with NBS (non-bloody stool). All O157:H7 isolates were *ehxA* positive and the majority (45/65, 69.23%) were also *stx*_2a_ + *stx*_2c_ positive. The majority of the *eae*-positive STEC isolates (166 of 173, 95.95%) carried *ehxA*. *ehxA* was overrepresented in isolates that carried *stx*_2a_ + *stx*_2c_ (*p* < 0.001) and *eae* (*p* < 0.001). The presence of *ehxA* was significantly associated with BD and O157:H7 (Table 1). However, no association was observed between the presence of *ehxA* and the duration of bacterial shedding, the age of the patients, or HUS (Appendix A).

### 2.2. Diversity of ehxA and Its Correlation to Serotypes and the stx and eae Genes

Thirty unique *ehxA* sequences (genotypes GT1 to GT30) were identified among the 199 *ehxA* positive STEC isolates. The nucleotide similarities of *ehxA* gene sequences in this study ranged from 95.79 to 100%. Five distinct subtypes (A, B, C, D, F) were found, out of which, subtype C (76, 38.19%) was the most predominant subtype, followed by *ehxA* subtype B (65, 32.66%) and *ehxA* subtype A (29, 14.57%). In addition, subtype C showed greater genetic diversity than other subtypes. All isolates carrying subtypes B, C, or F were *eae* positive, with the exception of two *ehxA* subtype C isolates (Table 2). Interestingly, all *ehxA* subtype A and D isolates were *eae* negative and subtype B was only found in O157:H7 isolates (Table 2). Other *ehxA* subtypes were represented within different serotypes: *ehxA* subtype C was linked to O121:H19 and O26:H11 strains (Appendix A) and subtype F mainly belonged to O103:H2 isolates (Table 2). The presence of *ehxA* + *eae* (Table 3) and *ehxA* + *eae* + *stx*_2_ (Appendix A) was statistically associated with O157:H7 isolates. The presence of *stx*_1_ and its subtype *stx*_1a_ was statistically associated with *ehxA* subtype F, while the presence of *stx*_2_ and its subtype *stx*_2a_ + *stx*_2c_ was linked to *ehxA* subtype B (Table 4).

A neighbor-joining tree was generated using 30 unique *ehxA* sequences from this study and 26 reference *ehxA* sequences that were reported previously. Three phylogenetic groups were identified, namely, group Ⅰ (*ehxA* subtype A), where all isolates were *eae* negative; group Ⅱ (*ehxA* subtype B, C, F), where all isolates were *eae* positive; group Ⅲ (*ehxA* subtype D) containing only two *eae*-negative isolates (Figure 1).

### 2.3. ehxA Subtypes and Phylogenic Groups in Correlation with Clinical Variables and the Presence of eae

*ehxA* subtype B was overrepresented in BD- and HUS-associated isolates. Accordingly, *ehxA* group Ⅱ was statistically associated with BD and HUS. *ehxA* subtype A and *ehxA* group Ⅰ were statistically associated with NBS and individuals ≥10 years of age; *ehxA* subtype F and *ehxA* group Ⅱ were significantly linked to individuals <10 years of age; however, these differences had no statistical significance after Benjiamini–Hochberg corrections (Table 5). *ehxA* subtype B and *ehxA* group Ⅱ were statistically associated with O157:H7 strains (*p* < 0.001) (Table 5). No association was observed between the *ehxA* subtype/phylogenetic group and the duration of bacterial shedding (data not shown). The presence of *ehxA* + *eae* (Table 3) and *ehxA* + *eae* + *stx*_2_ (Appendix A) was statistically associated with HUS. In addition, the presence of *ehxA* + *eae* was overrepresented in isolates from individuals <10 years of age (Table 3).

## 3. Discussion

Shiga toxin and intimin have been widely investigated as vital virulence factors of STEC [21]. In addition, enterohemolysin (*ehxA*) has emerged as a possible marker for the identification of specific STEC strains, such as O26, O157, O145, and O103, which are highly related to severe clinical symptoms, including BD and HUS [13,17,22,23]. The presence of *ehxA* was shown to be a useful epidemiological marker for the presence of Stx [12,14]. However, the role of *ehxA* in STEC pathogenicity and the association between *ehxA* and other key STEC virulence factors, such as *stx* and *eae*, have not been fully elucidated. Here, we systematically investigated the prevalence of *ehxA* in human clinical STEC isolates in Sweden and analyzed the association between *ehxA* and clinical symptoms, as well as the bacterial features. We found that *ehxA* was present in 83.26% of all clinical STEC isolates in this study. The majority of the *eae*-positive isolates also carried the *ehxA* gene, while only 50% of the *eae*-negative isolates carried *ehxA*. This was in line with a previous study, where *ehxA* was divided into two major phylogenetic clusters based on the presence or absence of *eae* [24]. *ehxA* was also shown to have a strong link with *eae*-positive atypical EPEC strains that were isolated from cattle and sheep [17]. Notably, the presence of *ehxA* was statistically associated with O157:H7, *stx*_2a_ + *stx*_2c_, and BD, suggesting that *ehxA* could be included as a virulence marker in clinical diagnostics to predict highly pathogenic STEC strains.

The phylogeny analysis showed that *ehxA*-positive STEC isolates were divided into three groups, as described in a previous study [20]. *ehxA* subtypes B, C, and F were assigned to group Ⅱ, among which, 98.8% carried the *eae* gene. *ehxA* subtypes A and D were assigned to groups Ⅰ and Ⅲ, respectively, which were all *eae* negative. This was in concordance with other studies [12,17], indicating that *ehxA* subtypes B, C, and F were closely associated with *eae*-positive strains. *ehxA* subtype C was the most predominant among the five subtypes, in accordance with a previous study demonstrating that clinical isolates mainly carried *ehxA* subtype C [12]. Importantly, we found that all O157:H7 isolates carried *ehxA* subtype B, which was consistent with a previous study [12]. This may also explain the associations we observed between *ehx**A* subtype B and *stx*_2_, BD, and HUS, since O157:H7 strains often carry *stx*_2_ and are highly associated with BD and HUS [19]. In addition, these results suggest that *ehxA* subtype B could indicate a higher virulence than other subtypes. Correspondingly, *ehxA* group Ⅱ, which included *ehxA* subtype B, was found to be statistically associated with O157:H7, BD, and HUS. *ehxA* group Ⅰ and subtype A was linked to NBS and individuals ≥10 years of age, while *ehxA* subtype F was associated with individuals <10 years of age, although these differences did not reach statistical significance after Benjiamini–Hochberg corrections. These data indicated that isolates belonging to *ehxA* group Ⅱ were highly pathogenic; *ehxA* phylogenic grouping could thus be used in the risk assessment of STEC infection.

Serotype O157:H7, *stx*_2_ subtype *stx*_2a_, and virulence genes *eae* and *ehxA* were often found to be more common in HUS patients [15,25,26]. The combination of *stx*_2_ and *eae* could increase the risk of developing severe clinical outcomes [27,28]. Here, we found that the presence of *ehxA* + *eae* and *ehxA* + *eae + stx*_2_ were statistically associated with HUS and O157:H7, indicating that the coexistence of more than one virulence factor could be associated with more severe clinical outcomes in STEC infections. In addition, the coexistence of *ehxA* and *eae* was linked to isolates from individuals <10 years of age.

There were some limitations in our study. First, we did not test the enterohemolytic phenotypes of the *ehxA*-positive clinical STEC isolates in this study. Previous studies showed that some *ehxA*-positive serotypes, for instance, O157:H^−^ [29] and O111:H^−^ [30], showed no enterohemolytic phenotype on washed sheep blood agar. Additional work is required to examine the enterohemolytic phenotype and its associations with the presence of the *ehxA* gene. Second, enterohemolysin can increase the level of the proinflammatory cytokine interleukin-1β in vitro studies [31]; additional studies are warranted to identify the role of enterohemolysin in STEC pathogenesis.

In conclusion, here we describe the prevalence and genetic diversity of *ehxA* gene in clinical STEC isolates collected in Sweden. Our results show that *ehxA* was prevalent in most of the clinical STEC isolates with high genetic diversity. We found that *ehxA* was often presented in *eae*-positive O157:H7 isolates and isolates from BD patients. Furthermore, *ehxA* subtype B and phylogenetic group Ⅱ were associated with severe clinical outcomes. Our study suggests that the coexistence of *ehxA*, *eae,* and *stx*_2_ could be used as a risk predictor for severe clinical symptoms in STEC infections.

## 4. Materials and Methods

### 4.1. Ethics Statement

The study was approved by the regional ethics committees in Gothenburg (2015/335-15) and Stockholm (2020-02338), Sweden.

### 4.2. Collection of STEC Isolates

All STEC isolates used in this study were described previously [20]. In total, 239 STEC isolates that were collected from STEC cases from 1994 to 2018 in Sweden were analyzed in this study (Appendix A). Bacterial genomic DNA was extracted and sequenced, as previously described [32]. The clinical picture was classified into HUS, BD, and individuals with non-bloody stool (NBS) [33]. Patients were divided into two age groups: <10 years and ≥10 years.

### 4.3. ehxA Subtyping and Polymorphism Analysis

The complete sequences of the *ehxA* gene were extracted from the genomic assemblies according to the genome annotation. The unique *ehxA* sequences were aligned with reference nucleotide sequences of the previously described *ehxA* subtypes (Appendix A). The sequences were aligned and the genetic distances between the *ehxA* subtypes were calculated using the maximum composite likelihood method with MEGA 7.0 software (Center for Evolutionary Medicine and Informatics, Tempe, AZ, USA), and a neighbor-joining tree was generated with 1000 bootstrap resamples. The *ehxA* genotypes based on *ehxA* sequence polymorphism was used to determine the diversity within each *ehxA* subtype.

### 4.4. Statistical Analyses

The associations between the *ehxA* prevalence or subtypes and bacterial features or clinical outcomes were analyzed using Fisher’s exact test. Statistica12 (StatSoft, Inc. Tibco, San Francisco, CA, USA) was used to determine the statistical significance, where *p*-value < 0.05 was considered statistically significant. Multiple testing corrections were done using the Benjamini–Hochberg method when needed.

## Figures and Tables

**Figure 1 toxins-13-00071-f001:**
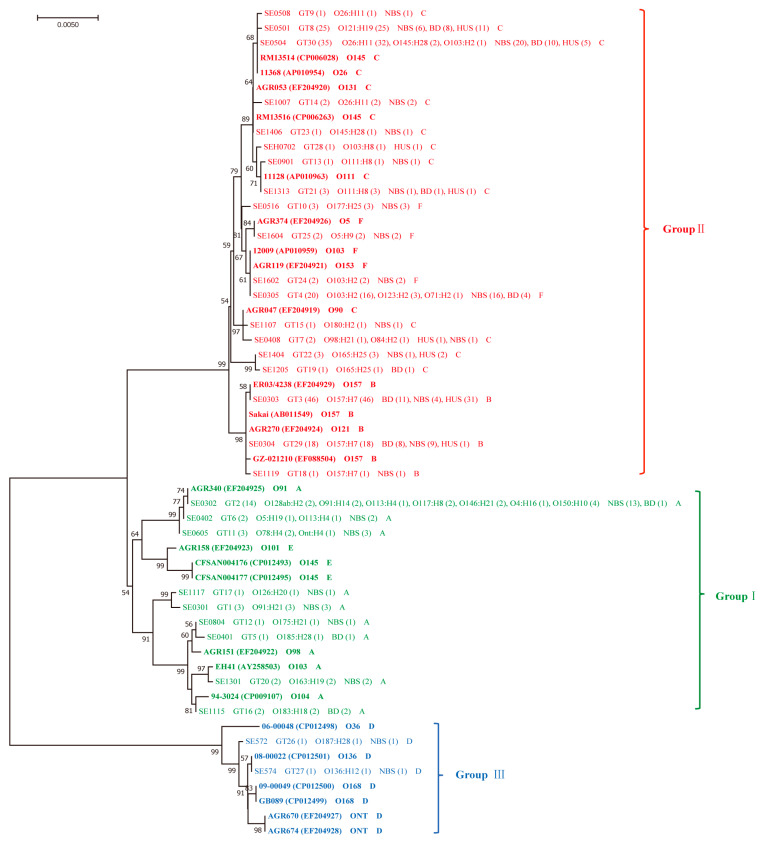
Phylogenetic relationships of the 30 unique *ehxA* sequences identified in this study and 26 reference sequences of six *ehxA* subtypes that were downloaded from GenBank based on the neighbor-joining method. The 26 reference sequences of six *ehxA* subtypes (A to F) are indicated in bold, the strain name of each reference sequence is shown, followed by the accession number in parentheses, the serotype, and the *ehxA* subtype. For the 30 unique *ehxA* sequences in this study, the representative isolate of each genotype is shown, followed by the corresponding *ehxA* genotype (number of isolates), serotypes (number of isolates), symptoms (number of isolates), and *ehxA* subtype. The phylogenetic groups of *ehxA* sequences are labeled in different colors. Bootstrap values above 50% are shown at the branch points. The scale bar indicates the genetic distance.

**Table 1 toxins-13-00071-t001:** Prevalence of the *ehxA* gene in 239 STEC isolates from Shiga-toxin-producing *E. coli* (STEC)-positive individuals ^a^.

*ehxA*	No. (%)		*p*-Value	No. (%)		*p*-Value	No. (%)		*p*-Value	No. (%)		*p*-Value
	BD (51)	NBS (128)		O157:H7 (65)	Non-O157 (174)		*stx*_2a_ + *stx*_2c_ (48)	Non-*stx*_2a_ + *stx*_2c_ (191)		*eae* + (173)	*eae* − (66)	
Positive	47 (92.16)	99 (77.34)	0.021 *	65 (100.00)	134 (77.01)	<0.001 *	48 (100.00)	151 (79.06)	<0.001 *	166 (95.95)	33 (50.00)	<0.001 *
Negative	4 (7.84)	29 (22.66)		0 (0.00)	40 (22.99)		0(0.00)	40 (20.94)		7(4.05)	33 (50.00)	

HUS: hemolytic uremic syndrome; BD: bloody diarrhea; NBS: non-bloody stool. ^a^ The associations were analyzed between the presence of *ehxA* and clinical symptoms (HUS and non-HUS; BD and NBS), age groups (<10 years of age; ≥10 years of age), duration of bacterial shedding (long: >24 days; short: ≤24 days), serotypes (O157 and Non-O157), *stx* subtypes, the presence of *eae*; only differences showing statistical significance were shown. * Statistically significant difference.

**Table 2 toxins-13-00071-t002:** Characteristics of *ehxA*-positive STEC isolates.

*ehxA* Subtype	No. of Isolates	Genotype (No.)	Group (No.)	*eae* (No.)		Serotype (No.)	*stx* Subtype (No.)	Symptoms (No.)	Age Group (No.)	Duration of Bacterial Shedding (No.)
				Positive	Negative					
A	29	GT1 (3), GT2 (14), GT5 (1), GT6 (2), GT11 (3), GT12 (1), GT16 (2), GT17 (1), GT20 (2)	Ⅰ (29)	0	29	O91:H21 (3), O113:H4 (2), O146:H21 (2), O150:H10 (4), O128ab:H2 (2), O4:H16 (1), O117:H8 (2), O91:H14 (2), O117:H8 (2), O185:H28 (1), O5:H19 (1), Ont:H4 (1), O78:H4 (2), O175:H21 (1), O183:H18 (2), O126:H20 (1), O163:H19 (2)	*stx*_2d_ (5), *stx*_2b_ (6), *stx*_1c_ (6), *stx*_2a_ (1), *stx*_1c_ + *stx*_2b_ (5), *stx*_1a_ + *stx*_2b_ (2), *stx*_1a_ + *stx*_2a_ (4), *stx*_1a_ + *stx*_2d_ (2)	NBS (25), BD (4)	<10 years (11), ≥10 years (18)	Short (7),long (9),NA (13)
B	65	GT3 (46), GT18 (1), GT29 (18)	Ⅱ (65)	65	0	O157:H7 (65)	*stx*_2a_ + *stx*_2c_ (45), *stx*_2c_ (9), *stx*_2a_ (2), *stx*_1a_ + *stx*_2c_ (9)	HUS (32), BD (19), NBS (14)	<10 years (18), ≥10 years (16), NA (31)	Short (17), long (8), NA (40)
C	76	GT7 (2), GT8 (25), GT9 (1), GT13 (1), GT14 (2), GT15 (1), GT19 (1), GT21 (3), GT22 (3), GT23 (1), GT28 (1), GT30 (35)	Ⅱ (76)	74	2	O84:H2 (1), O98:H21 (1), O121:H19 (25), O26:H11 (35), O111:H8 (4), O180:H2 (1), O165:H25 (4), O145:H28 (3), O103:H8 (1), O103:H2 (1)	*stx*_1a_ (36), *stx*_2a_ (33), *stx*_1a_ + *stx*_2a_ (4), *stx*_2a_ + *stx*_2c_ (3)	HUS (21), BD (20), NBS (35)	<10 years (36), ≥10 years (21), NA (19)	Short (21), long (19), NA (36)
D	2	GT26 (1), GT27 (1)	Ⅲ (2)	0	2	O187:H28 (1), O136:H12 (1)	*stx*_2g_ (1), *stx*_2a_ (1)	NBS (2)	<10 years (1), ≥10 years (1)	Short (2)
F	27	GT4 (20), GT10 (3), GT24 (2), GT25 (2)	Ⅱ (27)	27	0	O103:H2 (18), O123:H2 (3), O71:H2 (1), O177:H25 (3), O5:H9 (2)	*stx*_1a_ (24), *stx*_2c_ (24)	NBS (23), BD (4)	<10 years (21), ≥10 years (6)	Short (12), long (13), NA (2)

**Table 3 toxins-13-00071-t003:** Association between the presence of *ehxA* + *eae* and clinical symptoms, age groups, and serotypes.

*ehxA + eae*	No. (%)		*p*-Value	No. (%)		*p*-Value	No. (%)		*p*-Value
	Non-HUS (146)	HUS (53)		<10 Years of Age (87)	≥10 Years of Age (62)		O157:H7 (65)	Non-O157 (134)	
+	114 (78.08)	52 (98.11)	<0.001 *	75 (86.21)	42 (67.74)	<0.001 *	65 (100.00)	101 (75.37)	<0.001 *
−	32(21.92)	1 (1.89)		12 (13.79)	20 (32.26)		0 (0.00)	33 (24.63)	

* Statistically significant difference.

**Table 4 toxins-13-00071-t004:** Association between *stx* subtypes and *ehxA* subtypes ^a^.

*stx* Subtype	*ehxA* Subtype (No. Isolates)		*p*-Value	BH-Corrected *p*-Value
***stx_1_***	**F (27)**	**Non-F (172)**	<0.001 *	<0.001 *
+	24 (88.89)	42 (24.12)		
−	3 (11.11)	130 (75.58)		
***stx_1a_***	**F (27)**	**Non-F (172)**	<0.001 *	<0.001 *
+	24 (88.89)	36 (20.93)		
−	3 (11.11)	136 (79.07)		
***stx*** **_2_**	**B (65)**	**Non-B (134)**	<0.001 *	<0.001 *
+	56 (86.15)	51 (38.06)		
−	9 (13.85)	83 (61.94)		
***stx_2a_ + stx_2c_***	**B (65)**	**Non-B (134)**	<0.001 *	<0.001 *
+	45 (69.23)	3 (2.24)		
−	20 (30.77)	131 (97.76)		

^a^ The association was analyzed between the *stx* subtypes and *ehxA* subtypes; only differences showing statistical significance were shown. The *stx* subtypes and *ehxA* subtypes (number of isolates) were indicated in bold. * Statistically significant difference. BH: Benjamini–Hochberg.

**Table 5 toxins-13-00071-t005:** Association between the *ehxA* subtypes/groups and symptoms, age group, serotypes, and the presence of *eae*.

*ehxA* Subtype	No. Isolates	Symptoms												Age Group						Serotypes						*eae*					
		NBS (99)		BD (47)		*p*-Value	BH-Corrected *p*-Value	Non-HUS (146)		HUS (53)		*p*-Value	BH-Corrected *p*-Value	<10 Years (87)		≥10 Years (62)		*p*-Value	BH-Corrected *p*-Value	O157:H7 (65)		non-O157 (134)		*p*-Value	BH-Corrected *p*-Value	Positive (166)		Negative (33)		*p*-Value	BH-Corrected *p*-Value
		Pos	Prevalence	Pos	Prevalence			Pos	Prevalence	Pos	Prevalence			Pos	Prevalence	Pos	Prevalence			Pos	Prevalence	Pos	Prevalence			Pos	Prevalence	Pos	Prevalence		
A	29	25	25.25%	4	8.51%	0.018 *	0.535	29	19.86%	0	0.00%	<0.001 *	0.013 *	11	12.64%	18	29.03%	0.013 *	0.383	0	0.00%	29	21.64%	<0.001 *	0.002 *	0	0.00%	29	87.88%	<0.001 *	<0.001 *
B	33	14	14.14%	19	40.43%	<0.01 *	0.012 *	33	22.60%	32	60.38%	<0.001 *	<0.001 *	18	20.69%	16	25.81%	0.463	1.000	65	100.00%	0	0.00%	<0.001 *	<0.001 *	65	39.16%	0	0.00%	<0.001 *	<0.001 *
C	55	35	35.35%	20	42.55%	0.402	1.000	55	37.67%	21	39.62%	0.802	1.000	36	41.38%	21	33.87%	0.353	1.000	0	0.00%	76	56.72%	<0.001 *	<0.001 *	74	44.58%	2	6.06%	<0.001 *	0.001 *
D	2	2	2.02%	0	0.00%	0.327	1.000	2	1.37%	0	0.00%	0.392	1.000	1	50.00%	1	50.00%	0.809	1.000	0	0.00%	2	1.49%	0.32	1.000	0	0.00%	2	6.06%	0.027 *	0.810
F	27	23	23.23%	4	8.51%	0.032	0.969	27	18.49%	0	0.00%	<0.001 *	0.023 *	21	24.14%	6	9.68%	0.024 *	0.717	0	0.00%	27	20.15%	<0.001 *	0.003 *	27	16.27%	0	0.00%	0.013 *	0.381
***ehxA* Group**																															
Group Ⅰ	29	25	25.25%	4	8.51%	0.018 *	0.321	29	19.86%	0	0.00%	<0.001 *	0.008 *	11	12.64%	18	29.03%	0.013 *	0.230	0	0.00%	29	21.64%	<0.001 *	0.001 *	0	0.00%	29	87.88%	<0.001 *	<0.001 *
Group Ⅱ	115	72	72.73%	43	91.49%	<0.01 *	0.173	115	78.77%	53	100.00%	<0.001 *	0.005 *	75	86.21%	43	69.35%	0.012 *	0.225	65	100.00%	103	76.87%	<0.001 *	<0.001 *	166	100.00%	2	6.06%	<0.001 *	<0.001 *
Group Ⅲ	2	2	2.02%	0	0.00%	0.327	1.000	2	1.37%	0	0.00%	0.392	1.000	1	50.00%	1	50.00%	0.809	1.000	0	0.00%	2	1.49%	0.322	1.000	0	0.00%	2	6.06%	0.027 *	0.486

HUS: hemolytic uremic syndrome; BD: bloody diarrhea; NBS: non-bloody stool. Age groups (<10 years of age; ≥10 years of age). Pos: number of positive isolates. * Statistically significant difference.

## Data Availability

The sequences of all strains included in this study are openly available in GenBank with accession numbers and metadata shown in Appendix A.

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
