# Peer review of "Molecular Characterization of the Enterohemolysin Gene (ehxA) in Clinical Shiga Toxin-Producing Escherichia coli Isolates"

_toxins, 2021, doi:10.3390/toxins13010071_

Round 1

Reviewer 1 Report

The work entitled  “Molecular characterization of the enterohemolysin gene (ehxA) in clinical Shiga toxin-producing Escherichia coli isolates” is interesting and valuable.

Shiga toxin-producing E. coli strains are well known etiological factor of haemorrhagic colitis, haemolytic uremic syndrome bloody diarrhea, and thrombocytopenic purpura that are dangerous for humans. Due to the significant threat to human health and life in the event of STEC infection, it is very important to conduct monitoring studies with regard to these strains.

The aim of the study was to investigate the prevalence and genetic diversity of the ehxA gene, and its correlation with serotypes, presence of stx and eae. Moreover, the Authors assessed the association between ehxA subtypes and disease severity. The work brings a lot of new to our knowledge of the genetic diversity of STEC strains, which are an important etiological factor in severe human infections such as HUS and BD. The results obtained indicate that isolates belonging to ehxA group II are highly pathogenic and such grouping can be considered a good marker in estimating the risk of infection with STEC strains.

The results obtained in the current study are original, interesting and noteworthy.

Author Response

Thanks for the reviewer’s positive comments on our work.

Please see the attached revised manuscript.

Reviewer 2 Report

The aim of the research falls within the thematic scope of the journal. The purpose of this study was to determine the frequency and genetic diversity of the ehxA gene in Shiga toxin-producing E. coli, its correlation with serotypes, the presence of stx and eae genes, and to determine the possible relationship between the ehxA subtypes and the severity of the disease.

In the Introduction chapter, the Authors are weak on the principle that the names of toxins are not in italic as opposed to the names of their genes. Therefore, the text seems to be written in jargon. An example is the sentence: "At least 15 stx1 and stx2 subtypes have been identified, among which, stx2a, stx2c, and stx2d are more virulent than other subtypes as they are highly associated with severe clinical outcomes such as HUS [8]." (second sentence in the second paragraph of the Introduction). Whether the authors are talking about toxins or genes in this sentence? In terms of content, I have no objections, but the text would benefit if the Authors had spent some time "polishing" the style.

I have no objection to the chapters Results (apart from the comments on the Tables - below) and Discussion. Regarding chapter 4. Materials and Methods, the Authors refer to the publication from 2020 [20], which requires completion of the number (volume) and pages in the References chapter.

Tables 2, 3 and 5 should be done horizontally.

In the References section, in almost all items, journal names should be written in capital letters.

All comments were marked in the uploaded file in the review mode.

Author Response

Response to Reviewer 2 Comments

Comments and Suggestions for Authors

The aim of the research falls within the thematic scope of the journal. The purpose of this study was to determine the frequency and genetic diversity of the ehxA gene in Shiga toxin-producing E. coli, its correlation with serotypes, the presence of stx and eae genes, and to determine the possible relationship between the ehxA subtypes and the severity of the disease.

In the Introduction chapter, the Authors are weak on the principle that the names of toxins are not in italic as opposed to the names of their genes. Therefore, the text seems to be written in jargon. An example is the sentence: "At least 15 stxand stx2 subtypes have been identified, among which, stx2astx2c, and stx2d are more virulent than other subtypes as they are highly associated with severe clinical outcomes such as HUS [8]." (second sentence in the second paragraph of the Introduction). Whether the authors are talking about toxins or genes in this sentence? In terms of content, I have no objections, but the text would benefit if the Authors had spent some time "polishing" the style.

Response 1: We thank for this comment. All genes should be written in italic, when referring to toxins or protein, should not be in italic. We have checked and revised throughout the manuscript.

I have no objection to the chapters Results (apart from the comments on the Tables - below) and Discussion. Regarding chapter 4. Materials and Methods, the Authors refer to the publication from 2020 [20], which requires completion of the number (volume) and pages in the References chapter.

Response 2: We have completed the form of this reference as required (line 297).

Tables 2, 3 and 5 should be done horizontally.

Response 3: In our original submitted version, Table 2, 3 and 5 are actually presented horizontally, which might have been changed during editorial process. We have again reformatted the Table 2, 3, 4. We hope they will be presented horizontally in the revision.

In the References section, in almost all items, journal names should be written in capital letters.

Response 4: We have corrected these, please see the reference part (line 250-327).

Please see the attached revised manuscript.

Reviewer 3 Report

This paper presents a solid analyses of the correlation of the ehxA gene presence and its variants with various features of infections caused by Shiga toxin-producing Escherichia coli (STEC). This study adds a significant knowledge to our understanding of pathology of STEC strains. Some minor improvements of presentation are suggested below.

  1. Introduction - It is worth to mention (and perhaps to discuss very briefly) that stx genes are located in genomes of Shiga toxin-converting bacteriophages. 
  2. Introduction - The authors state that sequence of the ehxA gene consists of "about 2,997 base pairs". If the authors mean particular ehxA sequence, then "about" is unnecessary and even misleading. The sequence can be either exactly 2,997 bp long (if considering particular gene variant) or about 3 kb (if different variatns of this gene are considered).
  3. Introduction - In the last paragraph, indicate that data coming from ref. [18] are obtained from strains isolated in Sweden.
  4. Results - In the first sentence, mention that the isolates analysed in this paper were isolated in Sweden. Although this information is provided in Materials and Methods, it would be good to mention this shortly at the beginning of description of Results. 
  5. Results, subsection 2.2 - Please, describe shortly the level of diversity of the ehxA gene sequences. In fact, Table S5, presenting these sequences, is hard to follow without any allignment analysis.
  6. Large tables are hard to follow in their current format. Please, try to re-format them to avoid tranfers of parts of values to another line. Perhaps presentation of horizontal view of these tables might help. 

Author Response

Response to Reviewer 3 Comments

Comments and Suggestions for Authors

This paper presents a solid analyses of the correlation of the ehxA gene presence and its variants with various features of infections caused by Shiga toxin-producing Escherichia coli (STEC). This study adds a significant knowledge to our understanding of pathology of STEC strains. Some minor improvements of presentation are suggested below.

Introduction - It is worth to mention (and perhaps to discuss very briefly) that stx genes are located in genomes of Shiga toxin-converting bacteriophages. 

Response 1: We thank for this good suggestion. We have mentioned this in the revised manuscript (line 38-39).

Introduction - The authors state that sequence of the ehxA gene consists of "about 2,997 base pairs". If the authors mean particular ehxA sequence, then "about" is unnecessary and even misleading. The sequence can be either exactly 2,997 bp long (if considering particular gene variant) or about 3 kb (if different variatns of this gene are considered).

Response 2: The ehxA gene sequence length for most strains is 2,997 bp, a few are 2,994 bp or 2,928 bp. We have changed this to be about 3,000 base pairs (line 53).

Introduction - In the last paragraph, indicate that data coming from ref. [18] are obtained from strains isolated in Sweden.

Response 3: We have indicated this in the revised manuscript as suggested (line 65).

Results - In the first sentence, mention that the isolates analysed in this paper were isolated in Sweden. Although this information is provided in Materials and Methods, it would be good to mention this shortly at the beginning of description of Results. 

Response 4: We have mentioned this in the revision as suggested (line 75).

Results, subsection 2.2 - Please, describe shortly the level of diversity of the ehxA gene sequences. In fact, Table S5, presenting these sequences, is hard to follow without any allignment analysis.

Response 5: We added this statement ‘Nucleotide similarities of ehxA gene sequences in this study ranged from 95.79% to 100% in the revision (line 94-95).

Large tables are hard to follow in their current format. Please, try to re-format them to avoid tranfers of parts of values to another line. Perhaps presentation of horizontal view of these tables might help. 

Response 6: We have checked and re-formatted all tables in the revision. In our original submitted version, Table 2, 3 and 5 were actually presented horizontally, which might have been changed during editorial process. We are not sure if such changes will happen again in the revision, hope they will come in the horizontal forms.

Please see the attached revised manuscript.

Reviewer 4 Report

This is a clear and straightforward paper. The authors analysed the diversity and prevalence of the plasmid encoded enterohemolysin (EhxA) in a collection of 239 Shiga-toxin producing E. coli (STEC) strains isolated in Sweden that have been previously sequenced by Hua et collaborators (Hua et. al 2020, https://doi.org/10.1080/22221751.2020.1850182). They also analysed the association of this toxin with different clinical symptoms like bloody diarrhoea (BD) and haemolytic uremic syndrome (HUS).

By analysing the genome sequence of these 239 STEC strains they found that the ehxA was present in a majority of the collection. This gene was also associated with Bloody Diarrhoea (BD) but not with haemolytic uremic syndrome (HUS). They also identified a tight association of this toxin with O157:H7 isolates as well as with the presence of the eae gene and the shiga toxin alleles stx2a+stx2c. Up to 30 different ehxA variants were identified. These variants were classified within 5 different exhA subtypes previously described. A phylogenetic analysis of the 30 exhA identified variants, enriched with sequences retrieved from public databases, suggested the presence of three phylogenetic clusters. They also analysed the distribution of the different exhA variants and subtypes among different clinical variables, serotypes and the presence of the eae gene.  

This study does a good job on the analysis of the diversity of the ehxA gene in Sweden. They also analyse the association of the different variants of this gene with different E. coli serotypes, stx variants, and the presence of eae. However, there are several issues that must be addressed before the study is suitable for publication.

First of all, I want to mention that text lines in the manuscript version sent for evaluation were not numbered hindering the revision process.

The strains used in the study have been sequenced in a previous work (https://doi.org/10.1080/22221751.2020.1850182) and it does feel that the available data are underused. A phylogenetic inference of the strains may shed light on the nature of the association (convergent evolution? Or does it involve an event of Gene (or plasmid) acquisition in a common ancestor, with a posterior divergence?). Also the manuscript would also benefit from the identification (MobSuite may be useful for this analysis, https://github.com/phac-nml/mob-suite) of the large enterohemolysin-encoding plasmids. Are the same plasmids for each subtype/variants? (Not necessary an in-deep analysis)

Minor comments (since lines were not numbered I will refer to section number):

  • Section 2.1.

In the second phrase please write in parenthesis the meaning of NBS, it is the first appearance of this acronym.

Authors mention that the majority of O157:H17 are also stx2a+stx2c positive, write in parenthesis the proportion/numbers.

Table 1: The table is difficult to read; please try to keep the cell content in a single line, or at least avoid splitting words in two lines.

  • Section 2.2

Also how many of these unique sequences correspond to the 36 sequences described in the work of Fu and collaborators (https://doi.org/10.1038/s41598-018-22699-7.)

A multiple testing correction must be applied to the results presented in table 4.

Figure 1: It is difficult to read and interpret too many annotations on the leaves of the tree. Please change the figure tree editors like phandango (https://jameshadfield.github.io/phandango/#/) or Itol (https://itol.embl.de/) may be useful.

Table 4: p-values must be corrected for multiple tests.

  • Section 2.3

Table5: p-values presented on this table must be corrected for multiple tests.

Table5: It is difficult to read, please adjust better the cells.

  • Section 4.3

From the text (and from the methods section) I assume that the unique sequences were determined from nucleotide sequences, this in not clearly indicated in corresponding methods section.

  • Section 4.4

As mentioned in sections 2.2 and 2.3 in some analysis a multiple testing correction is required for some analysis.

Author Response

Response to Reviewer 4 Comments

Comments and Suggestions for Authors

This is a clear and straightforward paper. The authors analysed the diversity and prevalence of the plasmid encoded enterohemolysin (EhxA) in a collection of 239 Shiga-toxin producing E. coli (STEC) strains isolated in Sweden that have been previously sequenced by Hua et collaborators (Hua et. al 2020, https://doi.org/10.1080/22221751.2020.1850182). They also analysed the association of this toxin with different clinical symptoms like bloody diarrhoea (BD) and haemolytic uremic syndrome (HUS).

By analysing the genome sequence of these 239 STEC strains they found that the ehxA was present in a majority of the collection. This gene was also associated with Bloody Diarrhoea (BD) but not with haemolytic uremic syndrome (HUS). They also identified a tight association of this toxin with O157:H7 isolates as well as with the presence of the eae gene and the shiga toxin alleles stx2a+stx2c. Up to 30 different ehxA variants were identified. These variants were classified within 5 different exhA subtypes previously described. A phylogenetic analysis of the 30 exhA identified variants, enriched with sequences retrieved from public databases, suggested the presence of three phylogenetic clusters. They also analysed the distribution of the different ehxA variants and subtypes among different clinical variables, serotypes and the presence of the eae gene.  

This study does a good job on the analysis of the diversity of the ehxA gene in Sweden. They also analyse the association of the different variants of this gene with different E. coli serotypes, stx variants, and the presence of eae. However, there are several issues that must be addressed before the study is suitable for publication.

First of all, I want to mention that text lines in the manuscript version sent for evaluation were not numbered hindering the revision process.

Response 1: We apologize for this. But we cited the line numbers in our original submitted version, which might have been changed during editorial process. We are not sure if such changes will happen again in the revision, we have therefore uploaded a PDF format together with word format for the revised manuscript.

The strains used in the study have been sequenced in a previous work (https://doi.org/10.1080/22221751.2020.1850182) and it does feel that the available data are underused. A phylogenetic inference of the strains may shed light on the nature of the association (convergent evolution? Or does it involve an event of Gene (or plasmid) acquisition in a common ancestor, with a posterior divergence?). Also the manuscript would also benefit from the identification (MobSuite may be useful for this analysis, https://github.com/phac-nml/mob-suite) of the large enterohemolysin-encoding plasmids. Are the same plasmids for each subtype/variants? (Not necessary an in-deep analysis)

Response 2: We agree that there are several interesting questions could be further explored with the existing genomic data. We have actually performed genomic analysis of these strains to identify genetic factors associated with clinical outcomes, we did whole genome phylogeny to infer the evolution patterns, one relevant paper where we analysed part of strains collected from one region was recently accepted (https://www.frontiersin.org/articles/10.3389/fmicb.2021.627861/abstract), other manuscripts where we analysed strains from STEC-HUS patients are under review. Except the Shiga toxin (Stx) genes, we have not looked into diversity within other key genes and their correlations with disease outcomes, such data is very scarce and very interesting to know, we thus perform the current study, in attempt to understand the diversity of this specific enterohemolysin encoding-genes in clinical STEC strains and its clinical relevance.

We appreciate the review’s suggestion on the phylogenetic inference. We have therefore inferred the genealogical relationship of the 239 STEC isolates described in this study using whole-genome multi locus sequence typing (wgMLST), and plotted the exhA subtypes to the neighbour-net phylogenetic network that was generated from the analysis. Our preliminary results (see Figure S1) shown that the distribution patterns of different exhA subtypes could be rather different. For example, exhA subtype B was found only in isolates of a single cluster (O157 cluster). The subtype B genes appeared to be acquired by their common ancestor and transmitted vertically. The distributions of other subtypes can be both clustered and spread over distantly related lineages, suggesting both vertical and horizontal transmissions. We also totally agree with the reviewer’s suggestion on identification of large enterohemolysin-encoding plasmids, we are now planning to conduct long-reads sequencing to close the genomes of representative strains selected based on our previous, current, and ongoing studies. We will be able to extract complete genomes of plasmids, as well as prophages that we are interested to explore further, a comprehensive and thorough evolutionary and comparative analyses on the exhA-related plasmid victor is planned in our future study.

Figure S1. The neighbour-net phylogenetic network of the 239 STEC isolates was generated from whole-genome multi locus sequence typing (wgMLST) allele profiles of 2,390 loci that shared by their whole-genome assemblies. The presence of exhA subtypes in the genomes were marked in dots of different color.

Minor comments (since lines were not numbered I will refer to section number):

Section 2.1.

In the second phrase please write in parenthesis the meaning of NBS, it is the first appearance of this acronym.

Response 3: We have added the meaning of NBS in parenthesis (non-bloody stool) (line 78).

Authors mention that the majority of O157:H17 are also stx2a+stx2c positive, write in parenthesis the proportion/numbers.

Response 4: We have added the proportion/numbers in parenthesis (line 79).

Table 1: The table is difficult to read; please try to keep the cell content in a single line, or at least avoid splitting words in two lines.

Response 5: We have re-formatted tables to improve the readability.

Section 2.2

Also how many of these unique sequences correspond to the 36 sequences described in the work of Fu and collaborators (https://doi.org/10.1038/s41598-018-22699-7.)

Response 6: Ten unique ehxA sequences in our study were identical to 10 out of 36 unique ehxA sequences described in work of Fu, et al. The remaining 20 unique ehxA sequences in our study differ from sequences in Fu’s study.

A multiple testing correction must be applied to the results presented in table 4.

Response 7: We did the multiple testing correction by Benjamini-Hochberg correction, the adjusted p values were added in Table 4. We have updated the results (line 144-145) and discussion (line 197-198) accordingly.

Figure 1: It is difficult to read and interpret too many annotations on the leaves of the tree. Please change the figure tree editors like phandango (https://jameshadfield.github.io/phandango/#/) or Itol (https://itol.embl.de/) may be useful. 

Response 8: We agree and thank for this good suggestion. We have tried annotation using iTOL as reviewer suggested, it didn’t change much if we still keep all the information. We have therefore removed some information (e.g., stx subtypes, which can be found in Table S1), modified the Figure 1 as well as figure legend.

Table 4: p-values must be corrected for multiple tests.

Response 9: We did Benjamini-Hochberg correction, the B-H adjusted p values have been added in Table 4.

Section 2.3

Table5: p-values presented on this table must be corrected for multiple tests.

Response 10: We did Benjamini-Hochberg correction, the B-H adjusted p values have been added in Table 5.

Table5: It is difficult to read, please adjust better the cells.

Response 11: We have reformatted all tables in the revision for better readability.

Section 4.3

From the text (and from the methods section) I assume that the unique sequences were determined from nucleotide sequences, this in not clearly indicated in corresponding methods section.

Response 12: We have clarified this in the methods (line 237).

Section 4.4

As mentioned in sections 2.2 and 2.3 in some analysis a multiple testing correction is required for some analysis.

Response 13: We thank for this good suggestion, we have done Benjamini-Hochberg (BH) corrections for some analysis as suggested by reviewer, which have been updated in the method (line 247-248), results (line 144-145, Table 4 and Table 5) and discussion accordingly (line 197-198).

For Figure S1, sorry that it can't be uploaded in this review report. Will try other methods to upload it.

Please see the attached revised manuscript.

Round 2

Reviewer 4 Report

I think the manuscript is ready now for publication. Just one last comment, if the phylogenetic network finally appears as figure S1, there should be a section in the MM explaining how it was build.